# Next-generation diabetes diagnosis and personalized diet-activity management: A hybrid ensemble paradigm

Muhammad Sajid[1], Kaleem Razzaq Malik[1], Ali Haider Khan[2,3], Sajid Iqbal[4]*, Abdullah A. Alaulamie[4], Qazi Mudassar Ilyas[4]

1 Department of Computer Science, Air University, Islamabad, Pakistan, 2 Department of Software Engineering, Faculty of Computer Science, Lahore Garrison University, Lahore, Pakistan, 3 School of Software Engineering, Beijing University of Technology, Beijing, China, 4 Department of Information Systems, College of Computer Sciences and Information Technology (CCSIT), King Faisal University, Al-Ahsa, Kingdom of Saudi Arabia

* siqbal@kfu.edu.sa

**Data Availability Statement:** PIMA and Food datasets are publicly available and can be accessed using the following links: https://www.kaggle.com/datasets/uciml/pima-indians-diabetes-database

## Abstract

Diabetes, a chronic metabolic condition characterised by persistently high blood sugar levels, necessitates early detection to mitigate its risks. Inadequate dietary choices can contribute to various health complications, emphasising the importance of personalised nutrition interventions. However, real-time selection of diets tailored to individual nutritional needs is challenging because of the intricate nature of foods and the abundance of dietary sources. Because diabetes is a chronic condition, patients with this illness must choose a healthy diet. Patients with diabetes frequently need to visit their doctor and rely on expensive medications to manage their condition. It is challenging to purchase medication for chronic illnesses on a regular basis in underdeveloped nations. Motivated by this concept, we suggest a hybrid model that, rather than depending solely on medication to evade a visit to the doctor, can first anticipate diabetes and then suggest a diet and exercise regimen. This research proposes an optimized approach by harnessing machine learning classifiers, including Random Forest, Support Vector Machine, and XGBoost, to develop a robust framework for accurate diabetes prediction. The study addresses the difficulties in predicting diabetes precisely from limited labeled data and outliers in diabetes datasets. Furthermore, a thorough food and exercise recommender system is unveiled, offering individualized and health-conscious nutrition recommendations based on user preferences and medical information. Leveraging efficient learning and inference techniques, the study achieves a meager error rate of less than 30% using an extensive dataset comprising over 100 million user-rated foods. This research underscores the significance of integrating machine learning classifiers with personalized nutritional recommendations to enhance diabetes prediction and management. The proposed framework has substantial potential to facilitate early detection, provide tailored dietary guidance, and alleviate the economic burden associated with diabetes-related healthcare expenses.

and https://fdc.nal.usda.gov/download-datasets.html.

**Funding:** This work was supported by the Deanship of Scientific Research, Vice Presidency for Graduate Studies and Scientific Research, King Faisal University, Saudi Arabia (Grant No. KFU241647 to SI).

**Competing interests:** NO authors have competing interests.

## Introduction

One somewhat prevalent metabolic disorder is diabetes. Type 2 diabetes commonly first appears in middle age, though it can strike at any age. However, there are also reports of similar conditions in children. Several factors, including sedentary lifestyles, body weight, genetic predisposition, and eating habits bring on diabetes. Diabetes that is not treated can lead to hyperglycemia, which is defined by unusually high blood sugar levels. For patients to live longer and have a higher quality of life, early diabetes detection is crucial [1]. Among the main modifiable risk factors for preventing type 2 diabetes include being overweight or obese, not exercising, and eating a bad diet. The majority of countries state that their national policies aim for roughly 89% physical activity and a healthy diet. However, when money and implementation are taken into account, things change [2].

The pancreas, one of the body's most important organs, influences how sugar, protein, and fat are used for daily energy. When insulin levels are low or absent, blood glucose, or blood sugar, concentrations will be high. Excess sugar in the urine will be expelled; this medical condition is called diabetic mellitus [3]. Diabetes was a factor in 1.5 million fatalities in 2012. Over 2.2 million deaths were attributed to heart diseases and other reasons, most likely brought on by blood glucose levels that were not in the optimal range. Patients with prediabetes must live healthy lives and receive symptomatic therapy and early identification [4]. Determining a person's susceptibility to and predisposition toward a chronic illness such as diabetes is important. Early identification reduces the risk of more serious health conditions developing and lowers the cost of treating chronic disorders. To help physicians make better decisions about patient treatment in high-risk situations, precise deductions from quantifiable medical indicators are essential, particularly in emergencies where a patient may be unconscious or unable to communicate [5].

Certain foods are consumed in specific amounts as part of a healthy diet to meet nutritional needs [6]. The primary causes of health problems are inadequate food and an unrestricted habitual diet [7]. To obtain these nutrients, the optimum diet is well-balanced. However, the average person is still ignorant of the most common reasons for either an excess or a deficiency of minerals, such as calcium, proteins, and vitamins [8]. According to a World Health Organization (WHO) study, 347 million people worldwide are estimated to have diabetes. Except for breastfeeding, no diet gives the body all the essential nutrients it needs to stay healthy and perform its functions [9].

Patients with Diabetes Mellitus (DM) can be diagnosed by a doctor either manually or automatically. Advances in artificial intelligence (AI) and machine learning (ML) have increased the likelihood and efficacy of early illness detection and diagnosis compared to manual DM identification techniques [10]. Benefits include a reduction in the likelihood of human error and reduced work for medical personnel. Systems for computer-based decision support might be essential to effective treatment and precise diagnosis. Big data is produced by the DM field based on laboratory analysis, patient reports, treatment, medication, and follow-ups, among other things. Putting together all the required data by hand is difficult. The quality of the data organization has been damaged by inappropriate data management [11]. The goal of this work is to develop a hybrid model technique for classifying and recognizing diabetes. Fig 1 illustrates the process of detecting diabetes. In this study, a hybrid model for people with recognized diabetes is created using both machine learning and deep learning. Classifiers such as SVM, RF, etc. are trained to identify people with diabetes using a diabetes dataset. The deep learning algorithm R-BM for the recommender system subsequently receives this data and uses it to suggest a healthy diet to diabetes patients.

For this study, a hybrid strategy including two distinct models was created; the first one will identify whether a patient has diabetes. The methodology that follows will recommend a

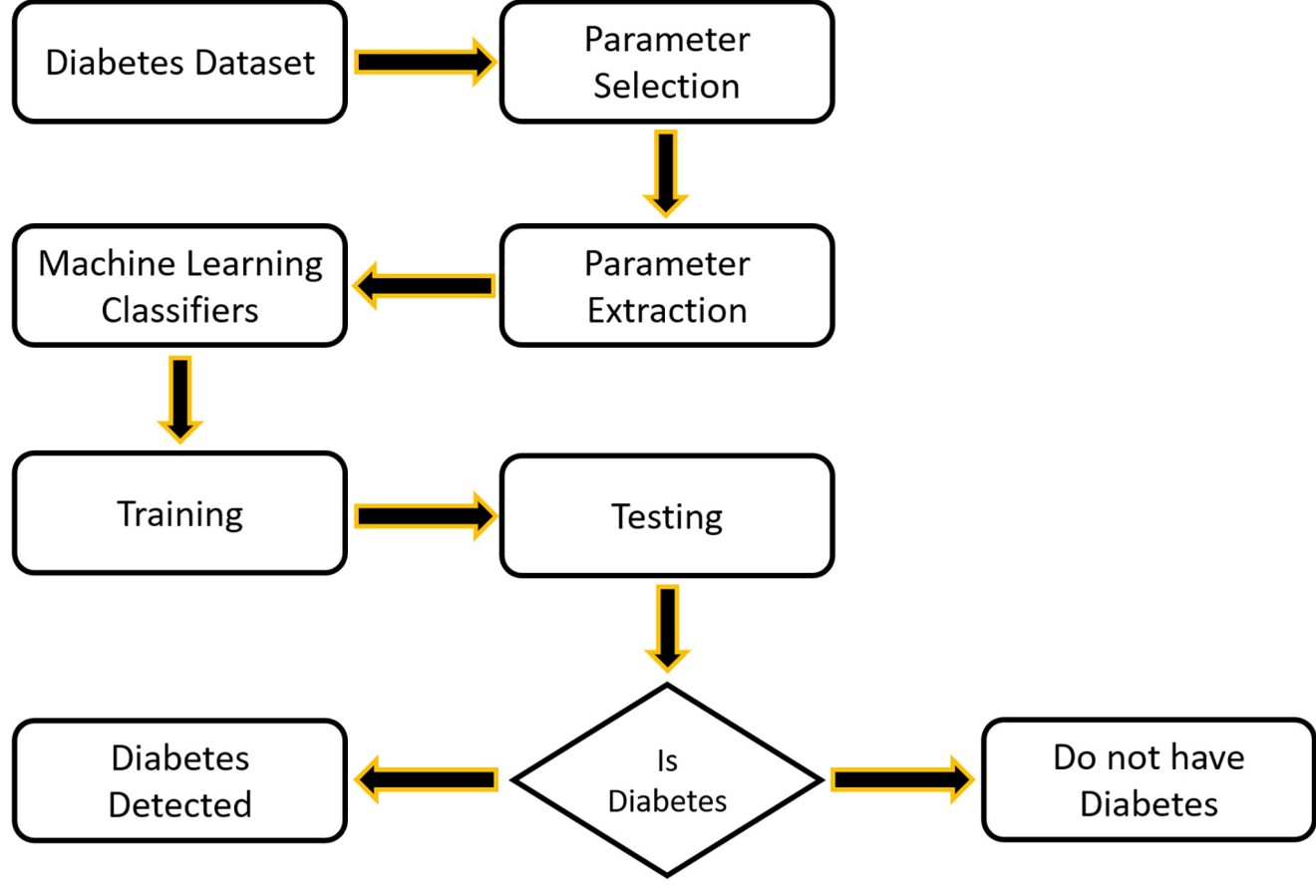

**Fig 1. Diabetes detection flow diagram.**

healthy diet for those with diabetes. Based on several risk factors, the system may determine a person's likelihood of developing diabetes, provide physicians with an early diabetes diagnosis, and inform patients of their doctor's advice for nutrition, exercise, and blood glucose monitoring. Finding the most significant feature in a dataset might be difficult. Given the contributions of several authors and researchers, even the best feature selection cannot guarantee significant quality, or 100% correctness [12]. Few researchers have developed a technique that accurately predicts instances using recurrent neural networks or deep learning.

The collaborative filtering strategy is used in this study. While collaborative filtering is a feasible approach, its ability to predict low numbers of likes is limited, which affects the ability to extract significant attributes. When developing deep learning models, restricted Boltzmann Machine (R-BM) approaches are essential since they yield more accurate results by hiding the details and exploiting quickly learned properties [13]. To address this problem, the optimal model parameters for root mean square error (RMSE) and recommended accuracy are selected. The following are a hybrid model's primary contributions:

1. This study offered a useful and optimal method for detecting diabetes in laboratory medical reports by identifying diabetic patients.

2. This study integrates machine learning classifiers for diabetes detection with Restricted Boltzmann Machine algorithms to suggest dietary and physical activity plans for people with diabetes.

3. Time and money can be saved by using the suggested strategy, which provides faster processing and computations.

4. In comparison to the current models for diabetes detection and dietary guidance, the suggested method is more condensed and contains fewer parameters. Test datasets were used to investigate it.

## Organization

The following outlines how this paper is organized: A related work is presented in the first section. Data screening, investigation, and data preprocessing are covered in the next section. Detailed work and discussion of the proposed hybrid model are given in the next section. The training section explains the training and testing process of the proposed model. The results of a hybrid model are presented in the next section. The study is concluded in the last section, with discussions of some future research.

## Related work

Foreseeing diabetes is crucial for effective treatment to prevent the disease's subsequent consequences. Studies on disease classification, diagnosis, prediction, and medicine have been done in great numbers. They have caused both conventional and machine learning techniques to advance and become noticeably more effective. For the classification of diseases, numerous ensemble methods and machine-learning algorithms have been employed [14]. Increased resistance to insulin in the body is a hallmark of type 2 diabetes, which means that the amount of insulin generated is insufficient to meet the body's metabolic requirements. The most typical kind of diabetes in people is type 2 diabetes [15].

This study suggested a model for diabetes prediction using an artificial neural network (ANN) to help doctors and other medical professionals. The following parameters were used in this investigation: a woman's age, the number of pregnancies, the Body Mass Index (BMI), heredity, height, and weight are all significant factors in determining whether or not a person develops diabetes [16]. The creation of these models is fraught with difficulties. T2D is a complex metabolic condition with a wide range of symptoms and concomitant diseases that are associated with it. For this condition to be controlled and for afflicted individuals to receive successful treatment, it is crucial to identify the key aspects. T2D has high development and medical expenditures, although numerous poorly understood risk factors exist. However, much development work has been done on classifying T2D using various computational methods [17].

This limits the applicability of models developed in high-income nations. Second, a community-based strategy drastically restricts the volume and complexity of data that community health professionals may gather. Because of this, many high-income country models rely on cutting-edge data [18]. It is impossible to overestimate the advantages of digital healthcare. Digital healthcare can improve accessibility, minimize costs, and improve the quality of health services while reducing service delivery inefficiencies. Primarily, digital healthcare may provide an inescapable and practical solution to the issues facing the global healthcare sector now facing [19].

Software tools and methods called recommender systems make suggestions to users. Recommendations can be for anything, including books, movies, news, and music tracks. The recommendation system generates suggestions by considering user interests and contextual data. Given the wealth of information available, it is imperative to eliminate useless information

[20]. Web, app, and SNS platform traffic have increased dramatically due to the growth and adoption of the Internet and smart devices. Additionally, these platforms are gathering more and more diverse data that may be used to determine consumer preferences. Notably, the use of SNS platforms by users allows for the collection of a variety of data, including information on the user's followers and their followers' followers, tweet data, and user-uploaded content. Additionally, the development of wearable sensors connected to smart devices makes it easier to collect various user-related medical data and exercise-related data [21].

There has been extensive discussion of security and privacy issues in various areas of computer science in the literature. Numerous difficulties relating to the growth of RS research have been brought up, and several studies covering these topics have documented a few security and privacy issues for users [22]. The best that this study knows, there isn't detailed work that carefully considers modern privacy and security issues. As a result, this paradigm offers a thorough investigation into a range of privacy and security issues, including trust, authentication, end-user privacy, damaging attacks, the fairness of RSs, their bias, their filter bubbles, and their ethics [22]. Healthcare informatics currently makes extensive use of machine learning (ML). Various crucial tools facilitate medical data analysis using ML [23]. Modern hospitals have the required monitoring and data-gathering technologies because data is incorporated into and stored in large information systems. According to researchers, the best method for assessing health-related data is machine learning (ML). In hospitals, proper diagnostic values must be loaded into a computer program with patient data to execute the learning algorithm. Clinical information is provided for an appropriate diagnosis. The past medical histories of treated patients automatically provide knowledge of the clinical diagnosis [24].

## Preprocessing

From Kaggle, the dataset was downloaded as given in Table 1. The datasets provided by the United States Department of Agriculture are scattered and insufficient. The data must be in good condition to process and send to the restricted Boltzmann machine, a deep learning neural network approach. Diseases that contribute significantly to a massive rise in glucose levels can be brought on by a shortage or excess of compositional substances, as shown in Fig 2.

Four datasets were collected for the Deep Learning Neural Network as shown in Table 2. A critical factor in merging Food datasets is the Food ID. Other characteristics have been eliminated since they are unnecessary [25] as shown in Fig 3.

Food Nutrient is another table that contains details about food nutrients, including their quantity, ID, and name. Other significant components of this table include the Food ID and Nutrient ID fields. The final table is nutritional. It includes information such as each nutrient's ID, name, and quantity. Using the preprocessing dataset technique, Branded Food is combined with the Food Dataset based on Food ID to create Merged Food. Any features that are not necessary are removed before combining the datasets. The Nutrient ID is also used to join the Food Nutrient and Nutrient databases to create the Merged dataset. Non-essential features are eliminated before combining the two datasets. Restricted Boltzmann Machine is then given the Final Food dataset based on the Global Food ID [25].

Fig 4 illustrates a hybrid model architecture where patients are first given diet and exercise recommendations using deep learning algorithms after their diabetes has been recognized

**Table 1. Diabetes dataset.**

| Number of Records | Pathological Report Features |
| --- | --- |
| 2000 | 7 |

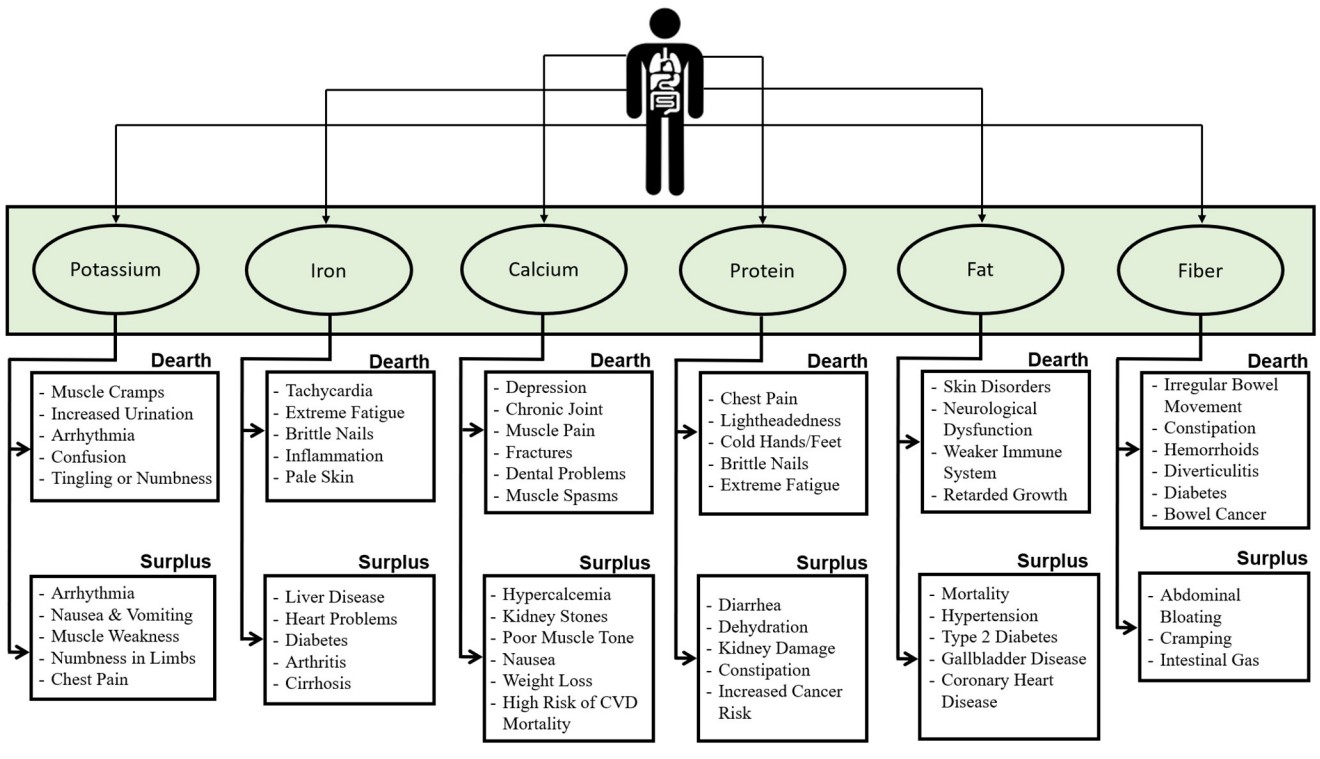

**Fig 2. Compositional elements and diabetes.**

using machine learning techniques. The human body needs specific components to function correctly. Two distinct datasets serve as inputs. The data is preprocessed, shown in Table 3, so the algorithm for calculating nutrition may use it. Preprocessing is completed to execute the nutrition calculation algorithm. The food inference algorithm requires the determination of the body's nutritional status to function correctly and suggest the right foods. The method for calculating nutrition uses input parameters. The nutrition calculation algorithm provides the recommended daily intakes of compositional ingredients depending on gender. The estimated nutritional value is then forwarded to the, which offers suggestions for food and exercise [26].

The RBM receives the preprocessed food dataset and calculates the diabetic patient's food intake. After ingesting the suggested foods, physical activity is also encouraged to ensure that the patient's body is kept in good condition and that they can live a happy life.

## Proposed hybrid model

Here is an illustration of a mathematical method that uses Random Forest to forecast diabetes [10]. Let Prob represent the likelihood that a patient has diabetes, and the input variables are $X\_1, X\_2, \ldots, X\_n$. Consequently, the Eq 1 can be used to represent the Random Forest model:

$$Prob = RF(X) \tag{1}$$

**Table 2. Diet and physical exercise dataset.**

| Unique Food Components | Input Features |
|---|---|
| 498182 | 4 |

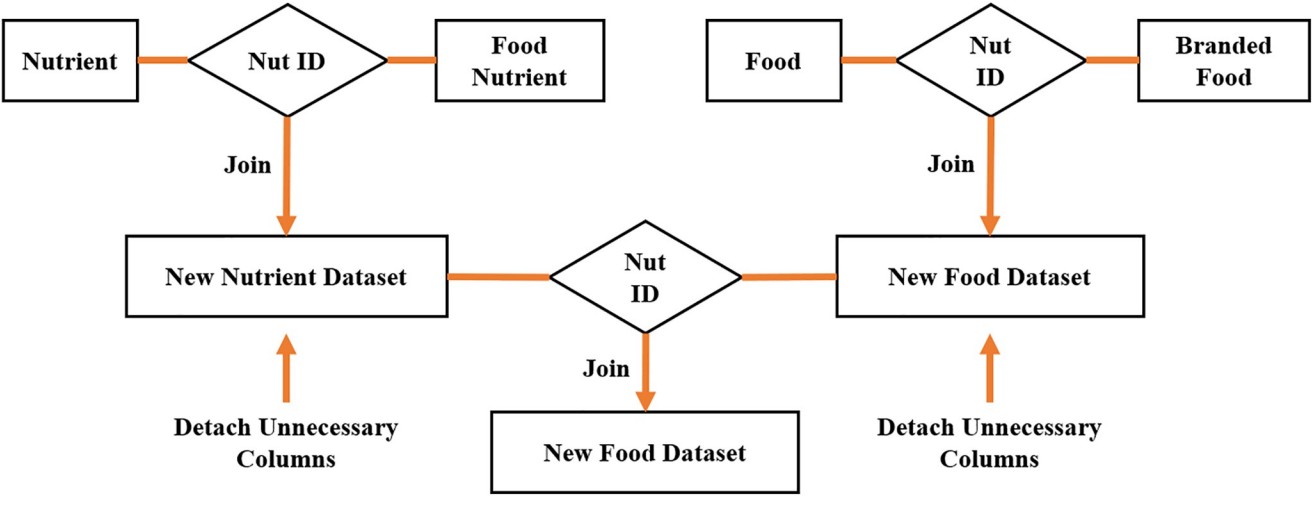

**Fig 3. Pre-processed dataset.**

**Fig 4. Architecture of hybrid model.**

**Table 3. Preprocessed dataset.**

| Patients | Foods | Features | Selected-Items |
|---|---|---|---|
| 500 | 1026179 | 30 | 7 |

RF stands for the Random Forest model, combining several different decision trees. Based on the majority vote of the decision trees, each decision tree $K_i$ in the RF contributes to the prediction through Eq 2:

$$Prob(K_i) = K_i(X) \tag{2}$$

The final probability is determined by Eq 3, which averages the probabilities of all the decision trees.

$$Prob = 1/n * SUM(Prob(K_i)) \tag{3}$$

Each decision tree receives the input variables X_1, X_2,. . ., and X_n and produces a binary decision based on a threshold value at each node. The decision tree's output is a binary classification of the patient as having diabetes. One of the main issues with C-filtering is the lack of data. As a result, those who have tasted the same dish are put together to train the RBM model. Each user group has its RBM. However, the amount of visible units fluctuates depending on how many everyday food items a group views or selects [10].

It is an apparent and concealed binary energy model as depicted in Fig 5. The weighted matrix W depicts the strength of connections between hidden layers $H_j$ and hidden layers $V_i$. Bias weights (offsets) are added for these units. The effectiveness of the RBM model can be improved by including additional hidden units. Visible nodes include Food Items, but hidden nodes have features that can be utilized to determine correlations between connected neurons. Therefore, the number of Visible-Units and Group-Size are influenced by the food item's attraction, as illustrated in Fig 6. RBMs determine the probability distribution for each food item before transferring it to hidden units that stand in for attributes. Scores for various scenarios will differ. This study graded numerous cases differently, as shown in Tables 4 and 5.

Some people might choose an inexpensive diet but detest the excessive protein in the score distribution. Conversely, Many patients could dislike eating affordable meals and choose highly-rated cuisine. As a result, several alternatives based on this study are implemented for various users. Positive weights are assigned to people associated with dislike, whereas negative consequences are given to people related to likes. The Model considers a user's history, popularity, and similarity scores and displays the options that are most appealing to the user. The similarities and likes ratings S, P $\epsilon$ 0,1 help with training when their values are near one. "similarity" refers to how a user distributes their ratings [27].

The following Eq 4 is used to determine how similar the food item "i" and user "u" are:

$$S_i(u, i) = \frac{\sum x \epsilon N_u Rating_{x,i}}{N(u)} \tag{4}$$

The following Eq 5 is used to determine how popular food item "i" is:

$$P_i(d, i) = \frac{\sum y \epsilon D_u Rating_{y,i}}{N(u)R_max} \tag{5}$$

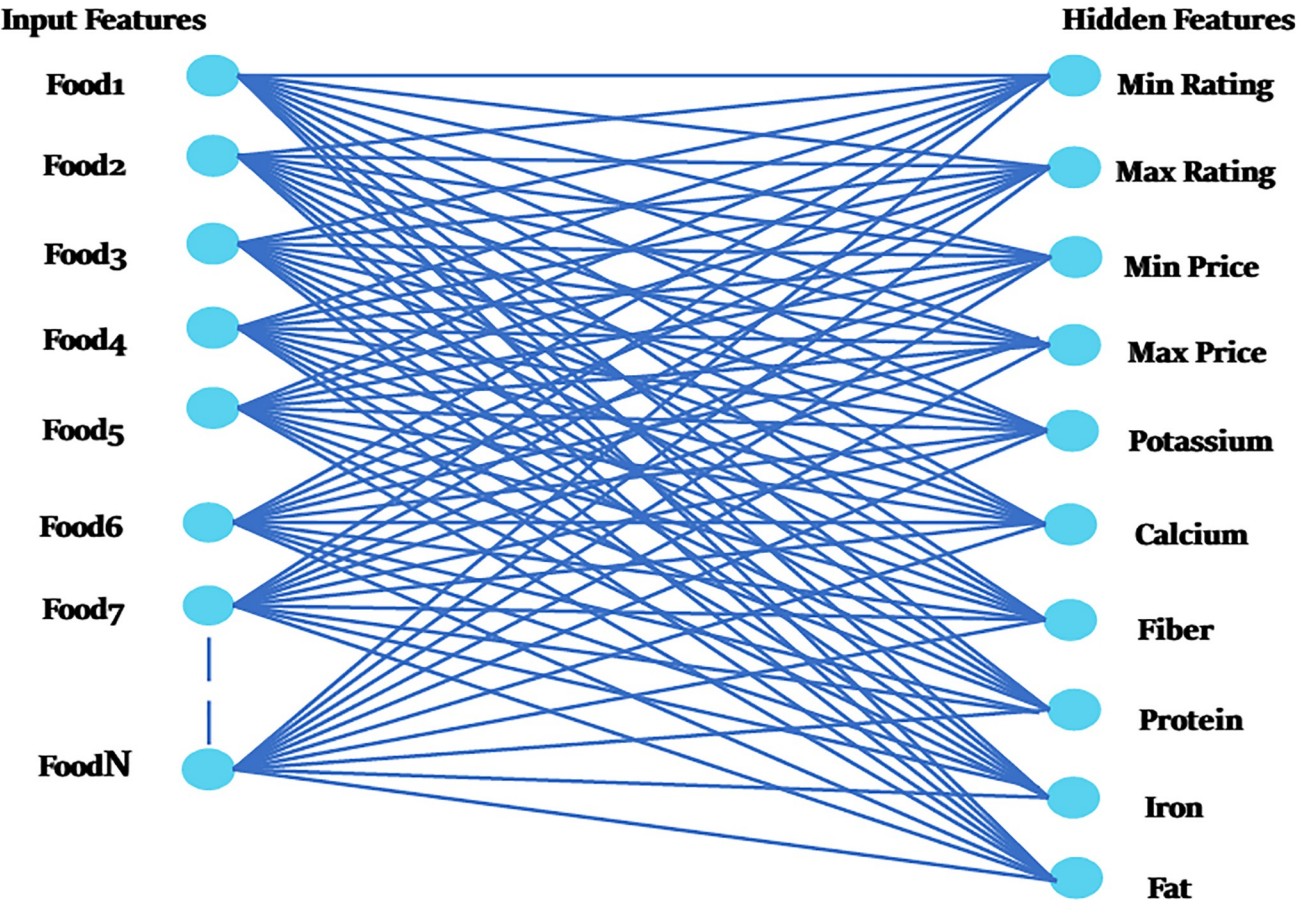

**Fig 5. Input and hidden features.**

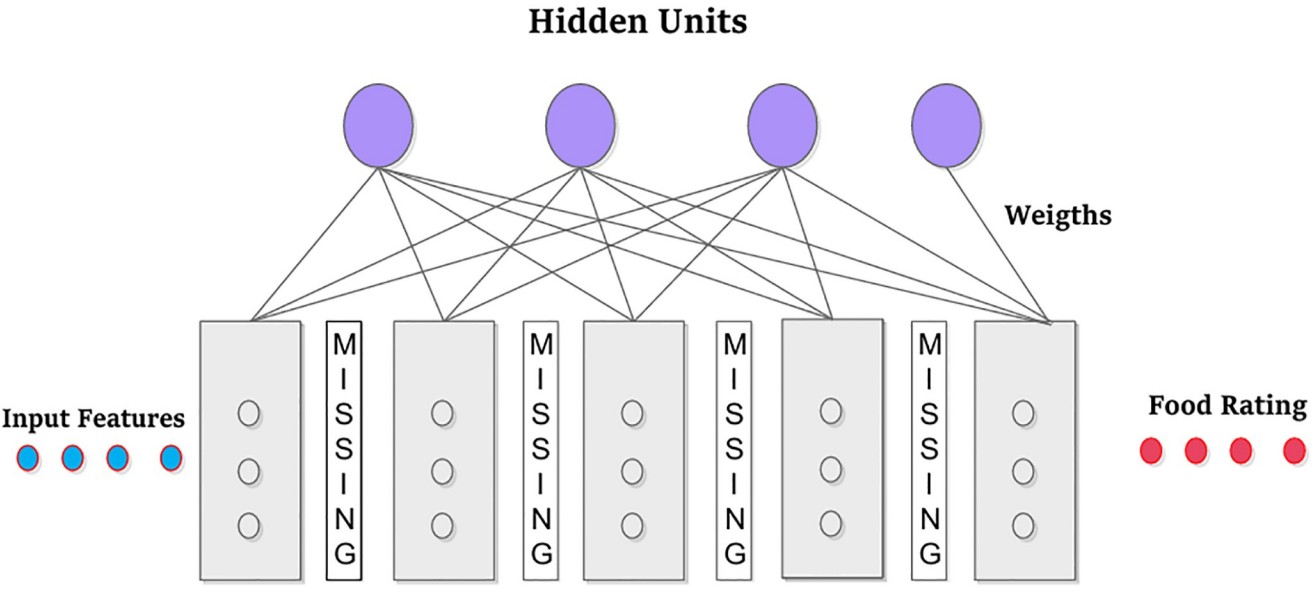

**Fig 6. Food-exercise RBM model.**

**Table 4. Different score distribution.**

| Patient ID | Min Rating | Max Rating | min Price | Max Price |
|---|---|---|---|---|
| 10 | 1 | 0 | 1 | 0 |
| 145 | 1 | 0 | 0 | 1 |
| 113 | 0 | 1 | 1 | 0 |
| 202 | 1 | 0 | 1 | 0 |
| 259 | 0 | 1 | 0 | 1 |
| 65 | 1 | 0 | 1 | 0 |
| 98 | 0 | 1 | 1 | 0 |
| 189 | 0 | 1 | 0 | 1 |

**Table 5. Different compositional elements score.**

| Patient ID | Potassium | Calcium | Fiber | Protein | Fat | Iron |
|---|---|---|---|---|---|---|
| 10 | 1 | 1 | 0 | 0 | 1 | 1 |
| 145 | 1 | 0 | 1 | 0 | 1 | 0 |
| 113 | 0 | 0 | 1 | 1 | 1 | 1 |
| 202 | 0 | 1 | 0 | 1 | 0 | 1 |
| 259 | 1 | 1 | 0 | 0 | 1 | 0 |
| 65 | 1 | 1 | 1 | 0 | 1 | 0 |
| 98 | 1 | 0 | 0 | 0 | 0 | 1 |
| 189 | 1 | 0 | 1 | 1 | 1 | 1 |

The energy is defined by the Equation below 6:

$$E = -\sum_i b_i v_i - \sum_i a_i h_i - \sum_i W_j v_i h_j \tag{6}$$

where the formula for activation energy is given by 7:

$$\alpha_i = \sum_j W_j x_j \tag{7}$$

W $ij$ is the weight between $i$ and $j$, and x$j$is the state of the unit (0 or 1). Fig 7 illustrates how groups of scenarios are created and rated according to the system's energy. While some scenarios obtain favorable evaluations, some do not, as seen in Fig 8.

High scores will correspond to probabilities likely to occur, whereas low scores will correspond to unlikely possibilities. Situations with high scores have heavier weights, whereas those with low ratings have lighter weights. In the model, zeros must not be used since there are some situations where the scores will be zeros. To reduce the number of zeros that could result in calculations going wrong and boost processing speed, exponential functions will convert positive and negative values into integers. Following exponential (Eq 8) and partition (Eq 9) functions are used to prevent zero scores [28].

$$Pr(vi, hi) = \frac{1}{k} e^{-E(vi,hi)} \tag{8}$$

where k is a technique for a division that may be described as:

$$k = \sum_{vi,hi} e^{-E(vi,hi)} \tag{9}$$

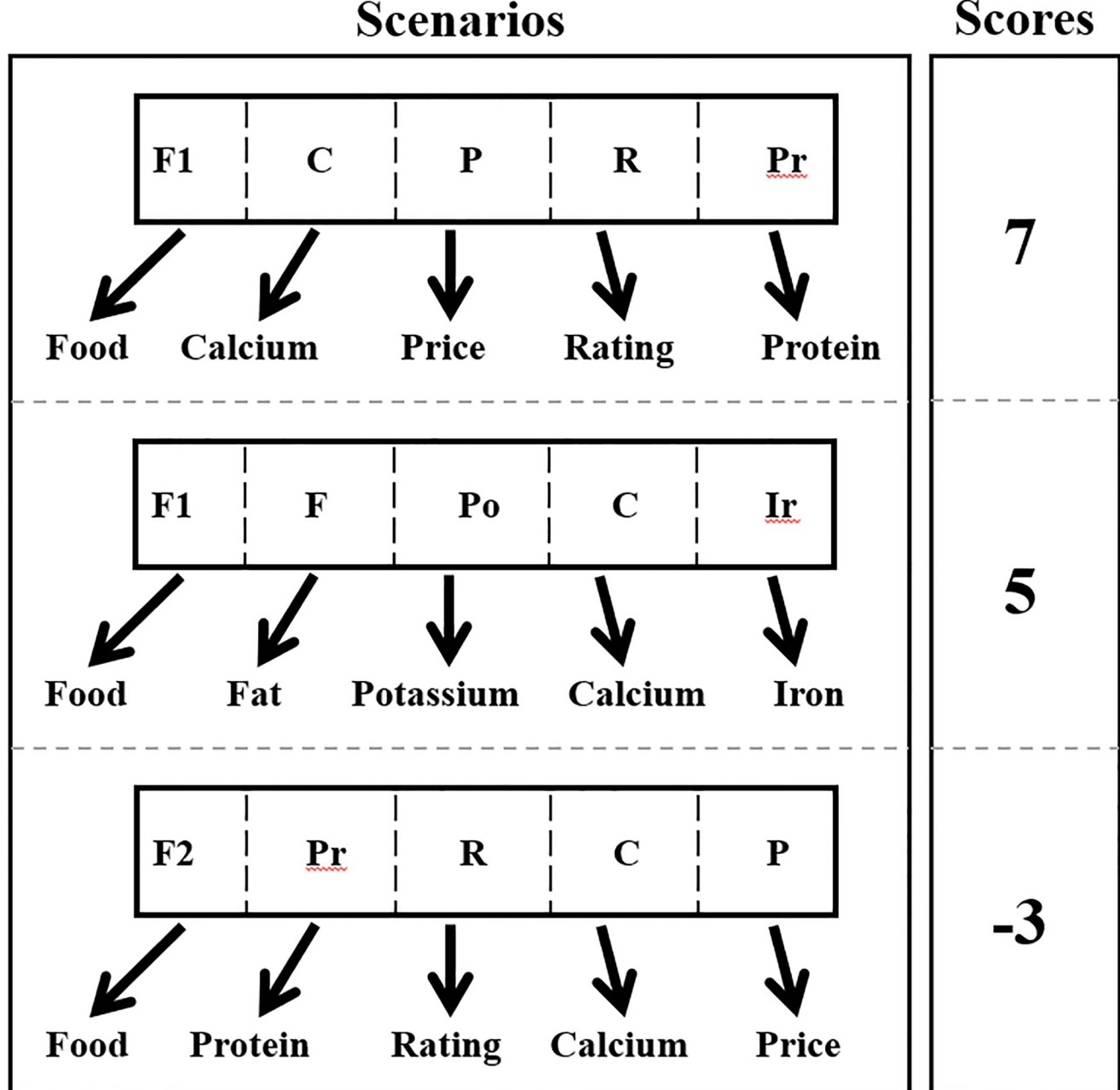

**Fig 7. Scenarios with scores assigned.**

The probability of every scenario is the same if weights are 0. In this case, all options would have a probability of 1, which is inconsistent with data because different scenarios call for different probabilities. Final probabilities must demonstrate that some food items score highly and others score poorly depending on whether they are inexpensive, well-liked by other patients, or provide more nutrients for that user. Contrastive divergence with restricted Boltzmann machines will increase the chances of various outcomes. The dataset's individual data points will all be evaluated. In the first instance, the scenarios involving highly rated food

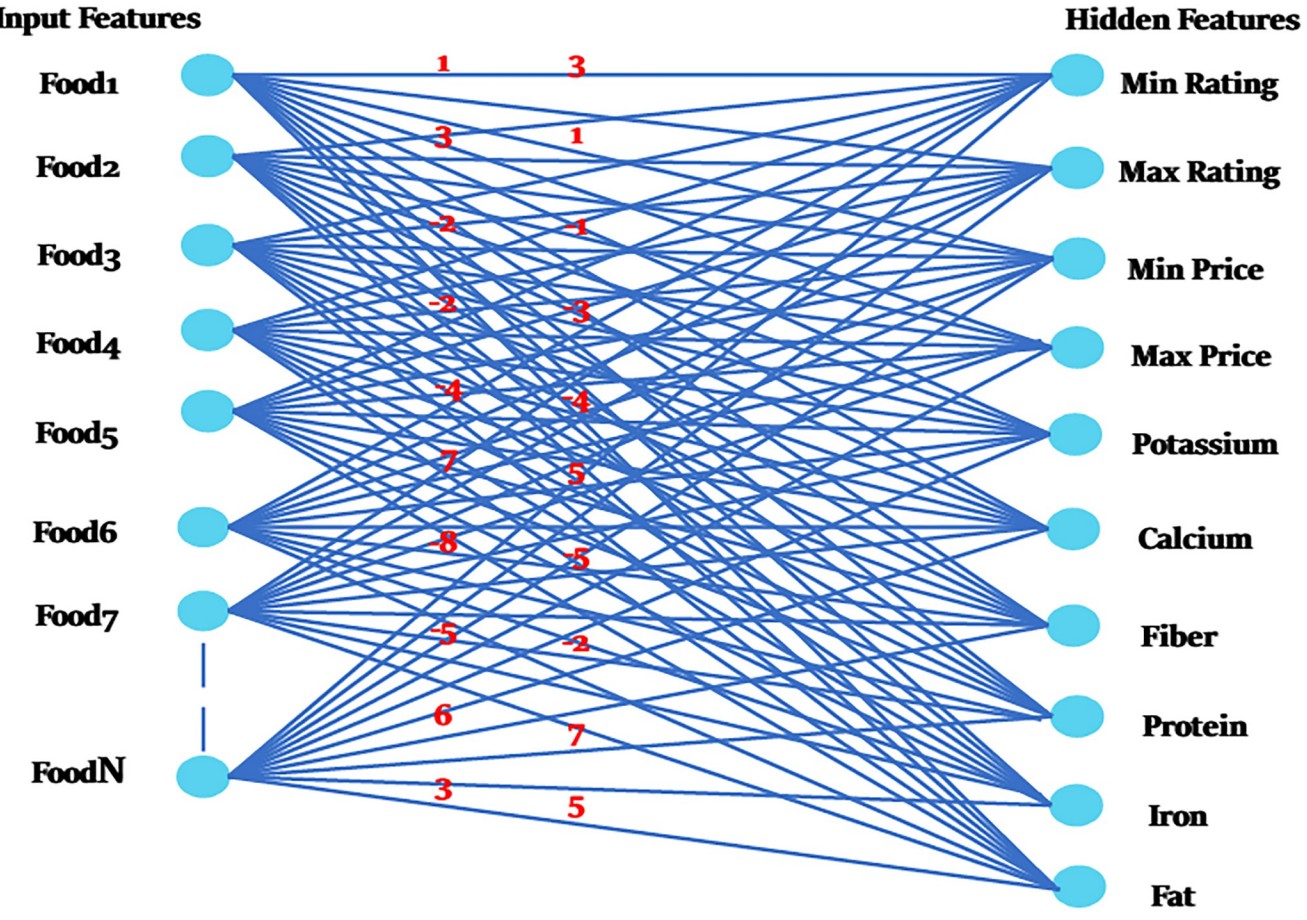

**Fig 8. Weightage assignment to different scenarios.**

products are emphasized, which slightly increases the probabilities while lowering the likelihood of all other possibilities. The chance of each food item having the highest ratings is enhanced in the dataset's following scenario, while the probability is decreased in all other cases. This process will be used for all datasets. As a result, foods are given high scores that assist in adjusting the weights. Eqs 10 and 13 are used to indicate the joint distribution of scores (V, H).

$$P(H_j = 1 | V, S, P) = \sigma(\alpha j + \sum_{i \epsilon V} V_i \cdot R_{ij} +$$

$$\sum_{i \epsilon S} S_i \cdot S_{ij} + \sum_{i \epsilon P} P_i \cdot T_{ij})$$

(10)

$$P(V_i = 1 | H) = \sigma(bi + \sum_{j \epsilon H} H_j \cdot R_{ij})$$

(11)

$$P(S_i = 1 | H) = \sigma(ci + \sum_{j \epsilon H} H_j \cdot S_{ij})$$

(12)

$$P(P_i = 1|H) = \sigma(di + \sum_{j \epsilon H} H_j \cdot T_{ij}) \tag{13}$$

where $\sigma$ is the activation function. The log-likelihood for training is found in the Eqs 14–16.

$$\triangle R_{ij} = \epsilon((V_i \cdot H_j)_{data} - (V_i \cdot H_j)_T) \tag{14}$$

$$\triangle S_{ij} = \epsilon((S_i \cdot H_j)_{data} - (S_i \cdot H_j)_T) \tag{15}$$

$$\triangle T_{ij} = \epsilon((P_i \cdot H_j)_{data} - (P_i \cdot H_j)_T) \tag{16}$$

where the rate of learning is $\epsilon$. The number of times that the features $i^{th}$ and $j^{th}$ of a food item are combined in user ratings is calculated using joint distribution [29]. This study employs the most often used gradient estimation, contrastive divergence [30]. The expected ratings for food item $q$ used for model testing are represented by the Eqs 17 and 18.

$$\bar{P} = P(H_j = 1|V, S, P) = \sigma(\alpha j + \sum_{i \epsilon V} V_i \cdot R_{ij} +$$

$$\sum_{i \epsilon S} S_i \cdot S_{ij} + \sum_{i \epsilon P} P_i \cdot T_{ij}) \tag{17}$$

$$P(V_q = 1|\bar{P}) = \sigma(aq + \sum_{j=1}^{F} \bar{P}jWqj) \tag{18}$$

There will be several calculations that call for more processing power. Gibbs sampling is a different restricted Boltzmann machine technique that has been discovered. By selecting data points randomly, Gibbs sampling can raise the likelihood of one particular event while lowering the likelihood of all other outcomes, as shown in Fig 9. A test with excellent results is

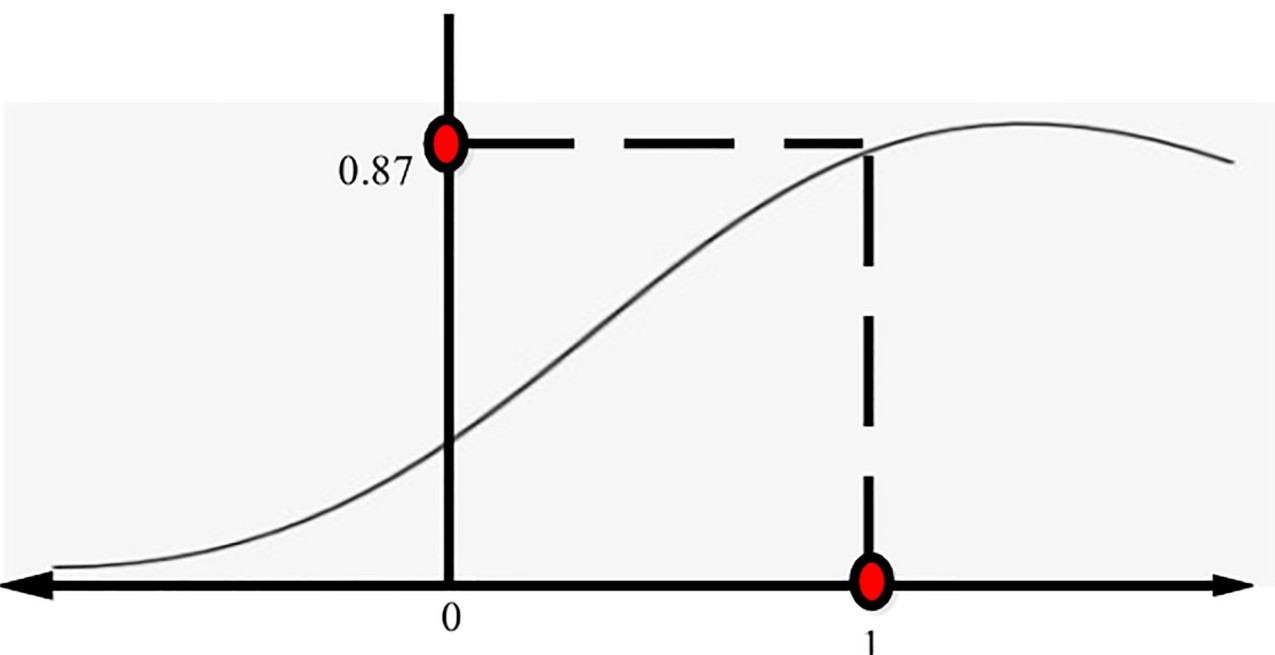

**Fig 9. Gibbs sampling & probability of individual random scenario.**

chosen randomly, increasing the likelihood of just that specimen. It is less likely that a sample with low scores will be the only one chosen at random.

This study uses a learning rate to lower the error rate. The training and testing of the sigmoid function with different numbers of Hidden Units are carried out. Having 20 concealed units yields the best results, as shown in the following graph. Experiments with a lot of concealed units have a larger RMSE. When updating sub-weight matrices, divide the learned gradient Eqs 19 and 20 by the group size to prevent learning rate volatility caused by grouping size [29]. The best outcomes are attained with a learning rate of 0.01. However, performance improves at 0.05 when learned weights from pre-trained models are used. This study uses two activation functions: [31].

$$SigmoidFunction : \sigma(x) = \frac{1}{1 + e^{-x}} \tag{19}$$

$$ReLUFunction : F(x) = \{0, x < 0 \, x, x \geq 0 \tag{20}$$

The root means square error (RMSE), accuracy, responsiveness (recall), and specificity are some of the metrics used to assess performance [32]. The RMSE in Eq 21 measures the discrepancy between estimated and actual values across samples:

$$RMSE = \sqrt{\frac{\sum_{i=1}^{N} (Y_i - \bar{Y}_i)^2}{N}} \tag{21}$$

where N is the size of the test set and $Y_i - \bar{Y}_i$ is the residual difference between the base and prediction values.

## Hybrid model training & testing

Fig 10 depicts a hybrid model workflow where the presence of diabetes in a patient is initially determined. If a patient has diabetes, a hybrid model feeds input data into the deep learning algorithm RBM for dietary and exercise advice. Table 6 displays an experimental setup with various parameters and their values for a hybrid model. The first phase of training an RBM with multiple parameters is shown. The number of nodes selected will correspond to the no. of rows in the weight matrix, while the number of hidden nodes will correspond to the number of columns. The first hidden node will get the vector multiplication of the inputs by the first column of weights before the matching bias term is added [33]. The RBMs communicate their learned consequences. Each RBM takes into account a sub-weight matrix. If a food item is not visible or selected, it is excluded from the weight matrix update. Fig 10 has been redrawn from the paper [7] to illustrate workflow for the hybrid model. It can be seen from Fig 10, that the model includes food items based on price and nutrition. The similarity and popularity of food products are defined by adding two additional layers, S and P. They display how well-liked particular foods are on menus where they appear and in earlier behaviors of similar individuals.

The visible levels in this model are unrelated, but the hidden layer (H) and the visible levels (V, S, and P) are entirely interconnected. Inputs are produced and rebuilt during the backward pass by translating numbers. The replication of the input data is taught to an RBM through numerous forward and backward passes [34].

An RBM is a feature extractor neural net family member, which is all about identifying data patterns. In training neural networks, one crucial hyperparameter is the learning rate. It establishes the increment size for updating the neural network's weights during training. The learning rate can significantly impact the neural network's performance and convergence. The dataset being utilized and the problem being addressed determine the learning rate. Finding

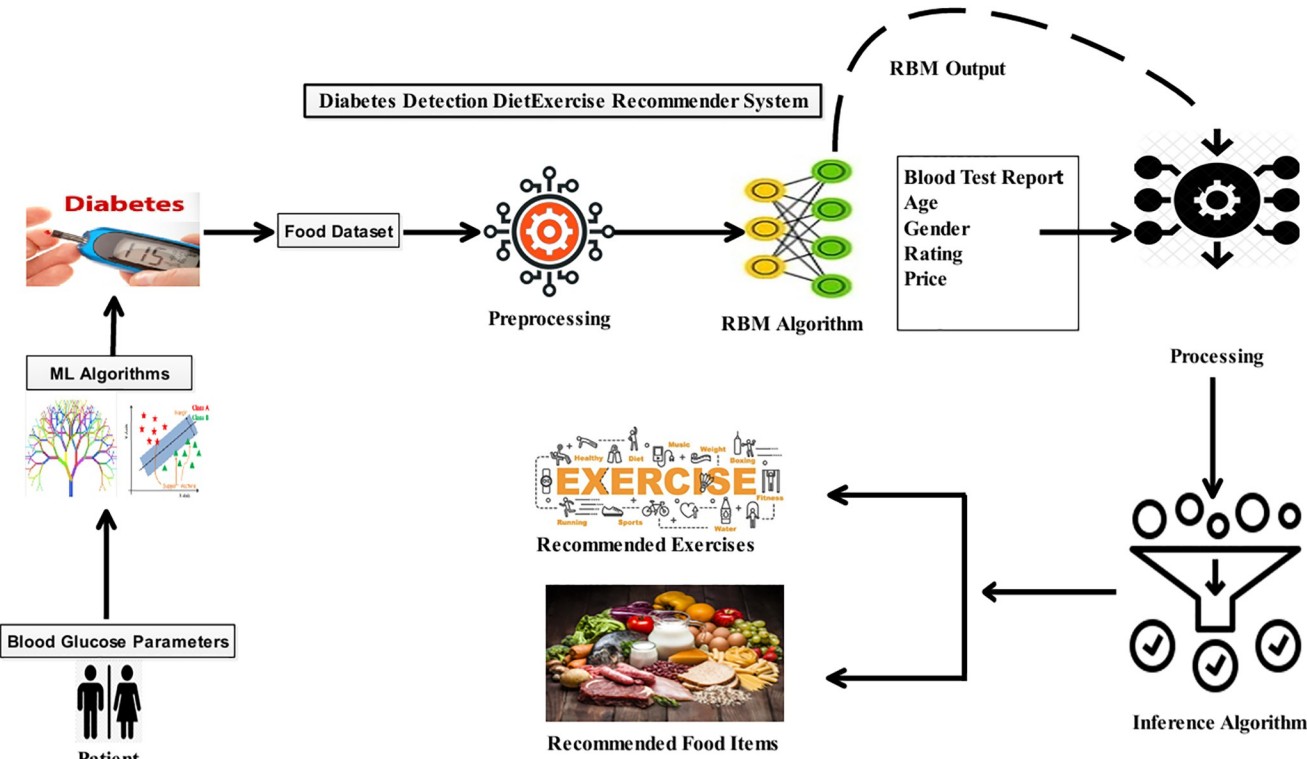

**Fig 10. Diabetes detection & diet exercise recommendation work flow.**

the ideal settings frequently requires testing out various learning rates. Strategies such as learning rate scheduling and adaptive learning rates can dynamically change the learning rate during training.

## Results

There are two stages to this research. First, this study determines whether a person has diabetes or not. This study suggests a diet and exercise recommender system for diabetic individuals at the following stage. The first stage of this work uses machine learning techniques to create a

**Table 6. Experimental setup.**

| Specification | Values |
|---|---|
| Total Food Items | 498182 |
| Number of Records | 2000 |
| Features in Pathological Reports | 7 |
| Compositional Elements | 250 |
| Pathological Reports | 100 |
| Number of Hidden Units | 1–200 |
| Number of Epochs | 10–70 |
| Platform | Jupiter Notebook |
| Languages | Python |
| Single Node System | RAM 8GB Intel Core i7 |

**Table 7. Experimental results.**

| Classifier | Accuracy | F1 Score | Recall | Precision |
|---|---|---|---|---|
| SVM | 0.78 | 0.77 | 0.77 | 0.77 |
| RF | 0.97 | 0.95 | 0.95 | 0.96 |
| Gradient Ada Boost | 0.85 | 0.86 | 0.83 | 0.89 |
| KNN | 0.86 | 0.78 | 0.78 | 0.78 |

rapid and accurate approach for detecting diabetes mellitus, which is essential for human health. If diabetes is not treated at the right time, it can lead to heart disease, blindness, stroke, renal failure, sexual dysfunction, lower-limb amputation, and issues during pregnancy in women.

Those who are overweight, physically inactive, or have a family history of the disease are at an increased risk of developing diabetes. It is critical to recognize diabetes in its early stages. Early diabetes mellitus diagnosis demands a new approach from earlier approaches. As a result, this study used a PIDD dataset and ensemble techniques such as Support Vector Machine, Gradient Boosting, and Random Forest.

Table 7 shows the results obtained from machine learning classifiers. RF is the best choice in this research when utilizing criteria like Accuracy, F1 Score, Recall, and Precision to evaluate classifiers. With the help of the PIDD dataset, many machine-learning classifiers were examined in this study, and Random Forest was chosen as the best classifier for detecting diabetes. The study uses deep learning to create a diet and exercise recommendation system for diabetic patients in the second stage, which involves identifying a patient as having diabetes. As the number of iterations rises, the model on the food dataset gets better—first without (V, S, P, and H layers), and then with (V, S, P, and H layers). We observed volatility in Root Mean Square Value (RMSE) starting at 0.70 and steadily decreasing to 0.63 after 50 epochs of iterations on our model with 50 hidden nodes and a learning rate of 0.01. We might need to try different combinations to get recommendations because this method does not provide us enough errors.

Our model underwent iterations with 200 hidden nodes and a learning rate of 0.01; then, after 50 epochs, the RMSE fluctuated, peaking at 0.68. If our approach does not provide us with a sufficient error as shown in Fig 11, we tried different combinations to get recommendations. [7]. Our model underwent iterations with 200 hidden nodes and a learning rate of 0.01; then, after 50 epochs, the RMSE fluctuated, peaking at 0.68. If our approach does not provide us with a sufficient mistake, we might have to try different combinations to get recommendations. This study employs a deep learning method with visible and hidden layers, followed by similarity, popularity, and visual and hidden layers. Both RBM models are trained and tested using the logistic sigmoid function, 100 hidden units, and a learning rate of 0.05. The R-BM model with Similarity $S$ and Popularity $P$ yields superior results because the layers (S and P) give the hidden layer additional information about the ratings via similarity and popularity scores. After performing iterations on the model, the volatility in RMSE Value started at 0.70 and decreased to 0.63, with less variance in RMSE after 50 epochs.

With 200 hidden nodes and a 0.01 learning rate, the model performs iterations, and the RMSE fluctuated throughout, peaking at 0.68 before dropping to 0.50 after 50 epochs. This study tested several combinations as shown in Fig 12 to produce recommendations if this strategy did not produce enough error. As a result, the error was reduced with various setups. The RMSE of the model with V, S, P, and H layers reduces to 0.50 after 50 epochs with 100 hidden nodes, as shown in Fig 13, and continues to decrease until it reaches 0.25, which the

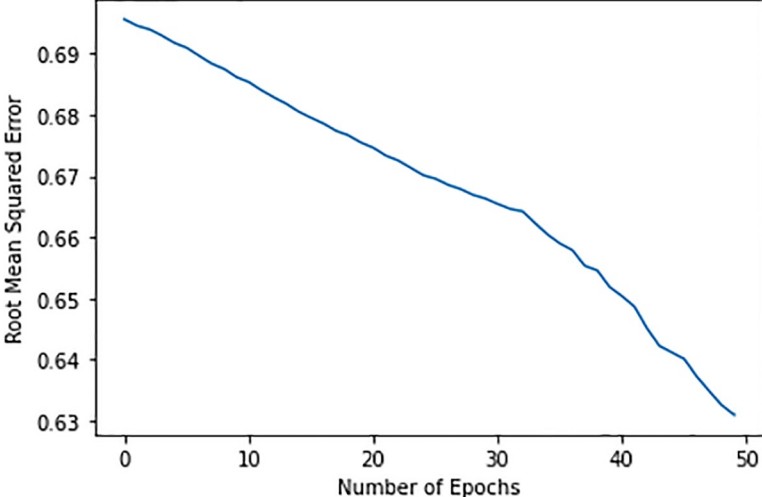

**Fig 11. Hidden-nodes 50 having learning-rate 0.01.**

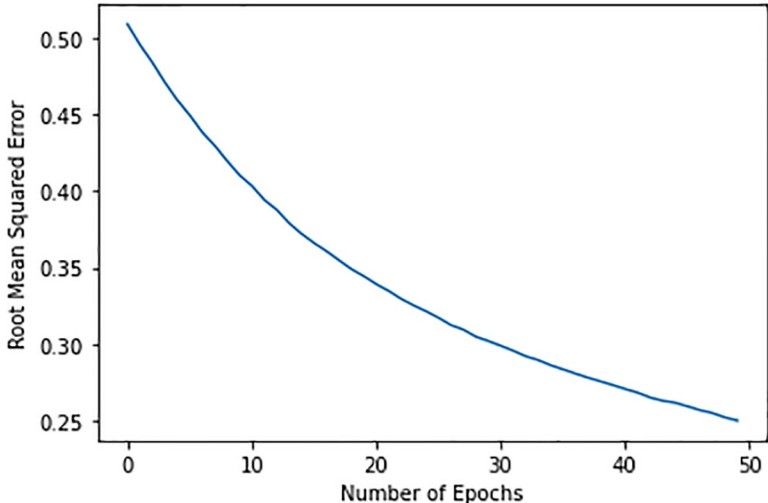

**Fig 12. Hidden-nodes 200 having learning-rate 0.01.**

desired outcome for the suggested model. Based on pathology data, we then utilize the model's output to suggest food and physical activity to diabetes patients. We format our recommendations for patients to see using OS, SYS, and colored libraries. The daily nutritional components for both males and females make up our dataset. We will enter the pathology results and use the daily intake to determine the necessary intake for the patients. The patient will be asked to enter their credentials before we provide recommended daily intakes, enabling them to determine how much daily consumption of food items containing nutritional elements they require.

Based on prediction accuracy, the proposed model creates a list of food items suggested for a user. This research provides a diet and exercise recommendation system that considers various input parameters [35]. The first pieces of data the recommender system needs are the

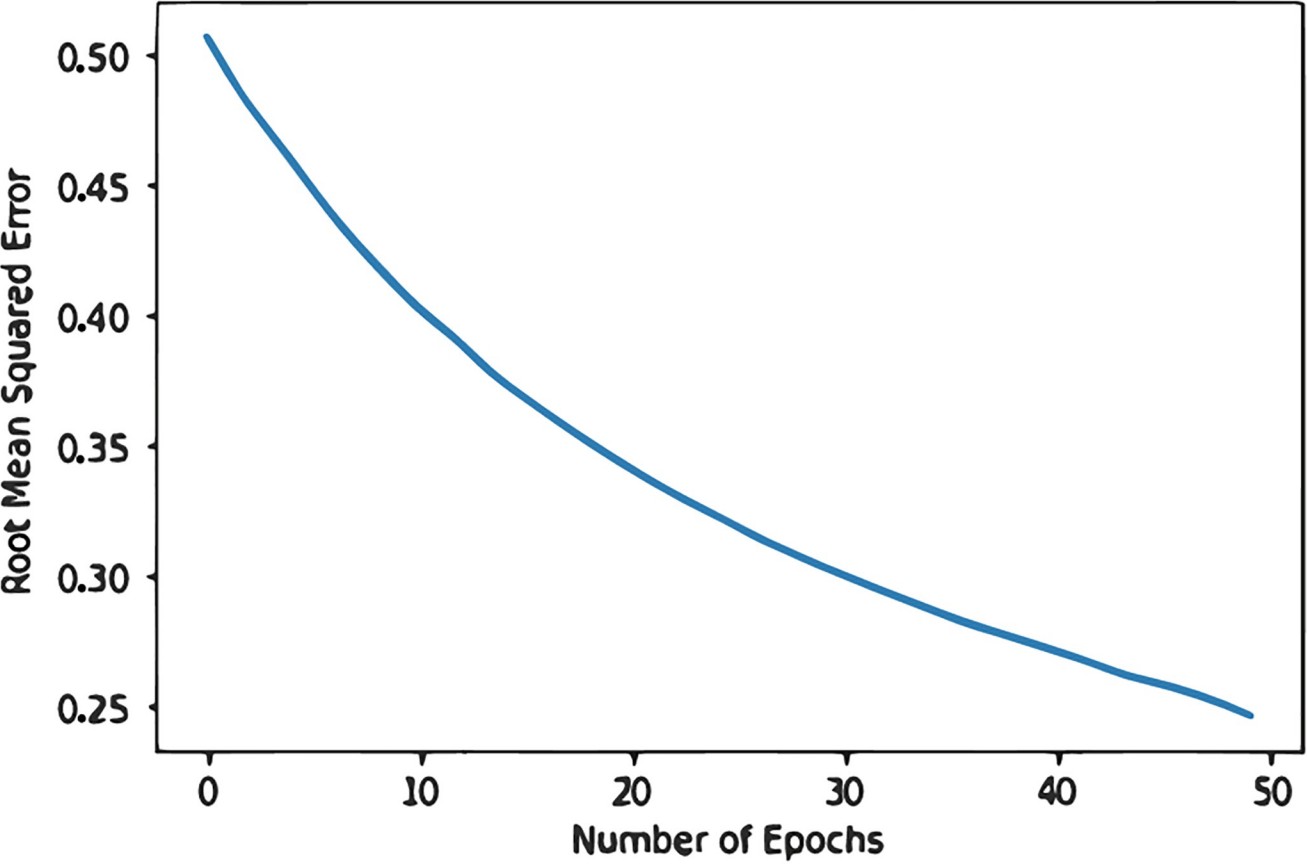

**Fig 13. Hidden-nodes 100 having learning-rate 0.05.**

patient's age, number, and pathology reports. Pathology data are used to determine the required nutritional value, and the patient is then asked to rate the food item based on user reviews. The final essential nutrition choice is then presented to the patient, and a pricing range is sought based on the suggested meals. Once processing has started, the algorithm will use the RBM reconstruction technique to provide the patient with a range of food options that will provide the necessary nutrients at a reasonable price [36]. After taking the necessary nutrients, the patient must engage in those physical activities to maintain exceptional health, a healthy lifestyle, and a physical appearance. The patient contributes to making society a healthier place to live [37]. Based on user preferences, the suggested model suggests 100 worth of food goods for each patient. There are some estimated diet and activity recommendations. Between 90 and 99 percent, there are about 40 food items per patient, and so on. The model's output is then kept in an Excel file. Tables 8 & 9 show the overall physical activity and diet recommendations as well as the recommendations for each patient.

After importing the suggested Excel file from the system and inputting input parameters, the inference algorithm will show food products based on reports and user preferences. This study presents the patient with advised food options, nutrient content, exercise, user preferences, and pricing. Finally, the suggested approach will show diet plans following user preferences and dietary constraints, but the inference technique will only display 10 food products with 100% suggestions. These patient preferences for choosing meals based on nutrition are then included in the preprocessed dataset for further use. The model generates meal

**Table 8. Recommendation of food-items.**

| No. of Iterations | 100% | 90% | 80% | 70% |
|---|---|---|---|---|
| 1 | 210 | 165 | 199 | 446 |
| 2 | 1301 | 1106 | 454 | 362 |
| 3 | 2070 | 260 | 314 | 461 |
| 4 | 2278 | 359 | 463 | 415 |
| 5 | 2489 | 760 | 591 | 1265 |
| 6 | 2114 | 185 | 284 | 340 |
| 7 | 2255 | 305 | 458 | 363 |
| 8 | 2375 | 198 | 358 | 201 |
| 9 | 2399 | 567 | 575 | 1180 |

**Table 9. Recommendation of diet and exercise per patient.**

| No. of Iterations | 100% | 90% | 80% | 70% |
|---|---|---|---|---|
| 1 | 55 | 19 | 26 | 21 |
| 2 | 190 | 49 | 41 | 35 |
| 3 | 286 | 69 | 62 | 45 |
| 4 | 236 | 71 | 69 | 67 |
| 5 | 328 | 87 | 77 | 81 |
| 6 | 180 | 27 | 37 | 39 |
| 7 | 233 | 38 | 46 | 41 |
| 8 | 294 | 59 | 50 | 48 |
| 9 | 370 | 60 | 67 | 55 |

suggestions using a larger dataset in a cyclical process. Then, this study uses those suggestions to make recommendations for both food and exercise.

## Discussion

To anticipate diabetes, this paper proposed classification models that can be used with electronic diagnostic devices implemented in hospitals. Using eight specified variables from the PIMA Indians dataset, the models were trained using three machine-learning techniques and assessed to determine whether a subject's diabetes mellitus diagnosis was positive. The experimental findings demonstrate that, when compared to the [38], the random forest classifier performed better on the entire Pima Indian Diabetes dataset than the SVM and other classifiers in terms of accuracy metric (97%), precision (96%), f-score (95%), and recall (95%). On the other hand, the random forest classification model beat the SVM and KNN models in terms of accuracy for the subsets that employed feature selection. On the PIMA Indians dataset, the KNN classifier outperformed the SVM model with an accuracy of 78% as opposed to 96%, which was the highest accuracy in this experiment. We may conclude that whilst a random forest performs better with more features, a random forest model performs better with a more precise feature selection for binary classification but struggles with many correlated features. Even though the models used in this experiment have accuracy levels close to 90% except for random forest which is 96%, our research's findings are consistent with earlier studies by [39, 40] and have room for improvement. The fact that our method did not exhibit overfitting was encouraging. Stated differently, the outcomes are more authentic and near to reality. The

current guidelines indicate that age and weight (indicated by BMI and skin fold thickness) are significant factors in the diagnosis and occurrence of diabetes mellitus.

Presently, recommender systems that rely on predictions are crucial in anticipating user behavior in their social interactions. It is difficult to forecast a user's behavior because of privacy issues and the scarcity of navigation records. To forecast and suggest diet and physical exercise to a diabetic user, we propose in this work a diet recommender model with joint distribution conditional on similarity and popularity scores [41]. Pre-training with the sample subset allowed for the identification of the ideal model parameters. According to experiments, 100 hidden units and a learning rate of 0.05 yield reasonably decent results. For the hidden layer, the sigmoid function yields the best accuracy. Because of the similarity and popularity scores, the hybrid model performs better than the clusters-based RBM models in terms of RMSE and accuracy, even if the PIMA dataset contains sparse data.

## Conclusion

Early detection of both DMs will make organizing prompt interventions easier and raise awareness of the disease's risk. After the detection of diabetes, the hybrid model recommends diet and exercise. A healthy diet is essential for those with a range of disorders. This paper describes a recommender system for diet and exercise that may provide customers with customized, healthy nutrition advice based on their preferences and pathological medical data. This study demonstrates how ratings of food products can be described using the Restricted-Boltzmann Machine. With more than 100 million in user-rated items, this study also indicates the practicality of employing RBMs to collect food data. The proposed research produces an error rate of less than $0.30\% in 50$ epochs using 100 hidden nodes. Instead of taking medication to avoid an expensive trip to the doctor, this study allows patients to eat nutrient-rich foods and practice preventive medicine. In the future, the model will be trained using several deep-learning methods. In the future, a user-friendly mobile and web interface will be offered to diabetic patients. This study aims to provide improvements on smart devices to give diabetic patients real-time dietary advice, leading to a healthier society.

## Supporting information

**S1 Data.**
(BST)

## Author Contributions

**Conceptualization:** Muhammad Sajid, Kaleem Razzaq Malik, Ali Haider Khan.

**Data curation:** Ali Haider Khan, Sajid Iqbal.

**Formal analysis:** Kaleem Razzaq Malik, Sajid Iqbal.

**Investigation:** Abdullah A. Alaulamie.

**Methodology:** Abdullah A. Alaulamie.

**Project administration:** Qazi Mudassar Ilyas.

**Resources:** Qazi Mudassar Ilyas.

**Supervision:** Qazi Mudassar Ilyas.

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
