## [Decision Letter · Decision Letter 0]

15 Mar 2024

PONE-D-24-07855Next-Generation Diabetes Diagnosis and Personalized Diet-Activity Management: A Hybrid Ensemble ParadigmPLOS ONE

Dear Dr. Iqbal,

Thank you for submitting your manuscript to PLOS ONE. After careful consideration, we feel that it has merit but does not fully meet PLOS ONE’s publication criteria as it currently stands. Therefore, we invite you to submit a revised version of the manuscript that addresses the points raised during the review process.

We look forward to receiving your revised manuscript.

Kind regards,

Suja A Alex, Ph.D.

Academic Editor

PLOS ONE

Journal Requirements:

3. Please be informed that funding information should not appear in the Acknowledgments section or other areas of your manuscript. We will only publish funding information present in the Funding Statement section of the online submission form. Please remove any funding-related text from the manuscript.

4. We note that Figure 10 in your submission contain copyrighted images. All PLOS content is published under the Creative Commons Attribution License (CC BY 4.0), which means that the manuscript, images, and Supporting Information files will be freely available online, and any third party is permitted to access, download, copy, distribute, and use these materials in any way, even commercially, with proper attribution. For more information, see our copyright guidelines: http://journals.plos.org/plosone/s/licenses-and-copyright.

a. You may seek permission from the original copyright holder of Figure 10 to publish the content specifically under the CC BY 4.0 license.

Reviewers' comments:

Reviewer's Responses to Questions

**Comments to the Author**

1. Is the manuscript technically sound, and do the data support the conclusions?

Reviewer #1: Yes

Reviewer #2: Yes

2. Has the statistical analysis been performed appropriately and rigorously? 

Reviewer #1: Yes

Reviewer #2: Yes

3. Have the authors made all data underlying the findings in their manuscript fully available?

Reviewer #1: Yes

Reviewer #2: Yes

4. Is the manuscript presented in an intelligible fashion and written in standard English?

Reviewer #1: Yes

Reviewer #2: Yes

5. Review Comments to the Author

Reviewer #1: Firstly, I would like to congratulate the authors for the excellent work so far, below are some suggestions to qualify the research, namely:

- Please explain in more detail the figures presented in the article, it highlights their importance for the present work;

- Finally, I suggest inserting one more topic, right after the Conclusion, addressing what is expected from this work in the future. This aspect is very important as it demonstrates the importance of the present study for the future;

Grateful.

Reviewer #2: Please find the following comments on your paper.

1. The motivation for research is not clear from the abstract.

2. The introduction section looks like a thesis writing format. Please change it to a scientific paper format. (Remove the problem statement and contribution heading from the introduction)

3. Please add a full stop at the end of the figure caption.

4. A state of art comparison is missing in the paper.

5. The result analysis should improve further to validate the approach.

6. It is unclear why the authors used Random Forest, Support Vector Machine, and XGBoost models for their study or excluded other ML models.

Regards

6. PLOS authors have the option to publish the peer review history of their article (what does this mean?). If published, this will include your full peer review and any attached files.

Reviewer #1: **Yes: **Gabriel Gomes de Oliveira

Reviewer #2: **Yes: **Alwin Poulose

---

## [Author Response · Author response to Decision Letter 0]

1 Apr 2024

Reviewer Response to Comments

1. The motivation for research is not clear from the abstract.

Author Response: Thanks for the comment. We agree with your remarks. The motivation for research has been included in the abstract and it is highlighted.

2. The introduction section looks like a thesis writing format. Please change it to a scientific paper format. (Remove the problem statement and contribution heading from the introduction)

Author Response: Thanks for the comment. We agree with your remarks. The writing format in the introduction section has been changed into a scientific paper format. Problem Statement and Contribution headings both are removed from the introduction. Page 1-3 are highlighted with the required change.

3. Please add a full stop at the end of the figure caption.

Author Response: Thanks for the comment. We agree with your remarks. We have added full stop at the end of each figure caption and highlighted the full stop. 

4. A state of art comparison is missing in the paper.

Author Response: Thanks you your valuable comments. A state of art comparison i

5. The result analysis should improve further to validate the approach.

Author Response: Thank you for valuable suggestion to improve the result analysis. We have added 2 more Figures 11-12 with different hidden units and learning rate and added some text to describe the figures and further improve the results on pages 14-16. All the changes are highlighted. 

6. It is unclear why the authors used Random Forest, Support Vector Machine, and XGBoost models for their study or excluded other ML models. 

Author Response: Thanks for the comment. We agree with your remarks. For diabetes prediction, Random Forest, Support Vector Machine (SVM), and XGBoost are often used models. These algorithms were selected because they have a high degree of accuracy in determining if a patient has diabetes or not. These techniques made managing complicated data easier and prevented the model from becoming overfit. In our dataset, these techniques offer versatility in handling various types of data. However, due to low performance necessitating more computation, our approach does not fit with other models such as KNN and Decision Tree. KNN uses a lot of memory and requires additional K values. In contrast, our model's decision tree is unstable and overfit. Not every problem can be solved by every algorithm. Only algorithms that can more accurately predict diabetes and perform better when training and testing the model with the provided dataset are the subject of this study. 

7. We note that Figure 10 in your submission contain copyrighted images. 

Author Response: Thanks for the comment. We agree with your remarks. Figure 10 is taken from My own Research Paper named RDED which shows workflow for hybrid model. Furthermore this figure has been redrawn and there is no copyright issue in Figure 10.

Few of the Screenshots are given here.

1. The motivation for research is not clear from the abstract.

2. The introduction section looks like a thesis writing format. Please change it to a scientific paper format. (Remove the problem statement and contribution heading from the introduction)

3. Please add a full stop at the end of the figure caption.

4. A state of art comparison is missing in the paper.

5. The result analysis should improve further to validate the approach.

---

## [Decision Letter · Decision Letter 1]

10 Jul 2024

Next-Generation Diabetes Diagnosis and Personalized Diet-Activity Management: A Hybrid Ensemble Paradigm

PONE-D-24-07855R1

Dear Dr. Iqbal,

We’re pleased to inform you that your manuscript has been judged scientifically suitable for publication and will be formally accepted for publication once it meets all outstanding technical requirements.

Kind regards,

M. Shamim Kaiser, PhD

Academic Editor

PLOS ONE

Additional Editor Comments (optional):

Reviewers' comments:

Reviewer's Responses to Questions

**Comments to the Author**

1. If the authors have adequately addressed your comments raised in a previous round of review and you feel that this manuscript is now acceptable for publication, you may indicate that here to bypass the “Comments to the Author” section, enter your conflict of interest statement in the “Confidential to Editor” section, and submit your "Accept" recommendation.

Reviewer #1: All comments have been addressed

Reviewer #2: All comments have been addressed

2. Is the manuscript technically sound, and do the data support the conclusions?

Reviewer #1: Yes

Reviewer #2: Yes

3. Has the statistical analysis been performed appropriately and rigorously? 

Reviewer #1: Yes

Reviewer #2: Yes

4. Have the authors made all data underlying the findings in their manuscript fully available?

Reviewer #1: Yes

Reviewer #2: Yes

5. Is the manuscript presented in an intelligible fashion and written in standard English?

Reviewer #1: Yes

Reviewer #2: Yes

6. Review Comments to the Author

Reviewer #1: Congratulations Dear Authors, after a long analysis it was observed that all corrections were successfully carried out. Therefore, I approve of the current work.

Grateful.

Reviewer #2: Dear Authors,

Thank you for addressing all my comments and I don't have any further comments about the paper.

Regards

7. PLOS authors have the option to publish the peer review history of their article (what does this mean?). If published, this will include your full peer review and any attached files.

Reviewer #1: **Yes: **Gabriel Gomes de Oliveira

Reviewer #2: No

---

## [Editor Report · Acceptance letter]

19 Oct 2024

PONE-D-24-07855R1 

PLOS ONE

Dear Dr. Iqbal, 

I'm pleased to inform you that your manuscript has been deemed suitable for publication in PLOS ONE. Congratulations! Your manuscript is now being handed over to our production team.

Kind regards, 

on behalf of

Dr. M. Shamim Kaiser 

Academic Editor

PLOS ONE